# The Potential for Natural Antioxidant Supplementation in the Early Stages of Neurodegenerative Disorders

**DOI:** 10.3390/ijms21072618

**Published:** 2020-04-09

**Authors:** Francesca Oppedisano, Jessica Maiuolo, Micaela Gliozzi, Vincenzo Musolino, Cristina Carresi, Saverio Nucera, Miriam Scicchitano, Federica Scarano, Francesca Bosco, Roberta Macrì, Stefano Ruga, Maria Caterina Zito, Ernesto Palma, Carolina Muscoli, Vincenzo Mollace

**Affiliations:** 1IRC-FSH Department of Health Sciences, University “Magna Græcia” of Catanzaro, Campus Universitario di Germaneto, 88100 Catanzaro, Italy; oppedisanof@libero.it (F.O.); jessicamaiuolo@virgilio.it (J.M.); micaela.gliozzi@gmail.com (M.G.); xabaras3@hotmail.com (V.M.); carresi@unicz.it (C.C.); saverio.nucera@hotmail.it (S.N.); miriam.scicchitano@hotmail.it (M.S.); federicascar87@gmail.com (F.S.); francescabosco@libero.it (F.B.); robertamacri85@gmail.com (R.M.); rugast1@gmail.com (S.R.); mariacaterina.zito@gmail.com (M.C.Z.); palma@unicz.it (E.P.); muscoli@unicz.it (C.M.); 2Nutramed S.c.a.r.l, Complesso Ninì Barbieri, Roccelletta di Borgia, 88021 Catanzaro, Italy; 3IRCCS San Raffaele, Via di Valcannuta 247, 00133 Rome, Italy

**Keywords:** neurodegeneration, oxidative metabolism, neuroinflammation, exogenous antioxidants, polyphenols, nutraceuticals

## Abstract

The neurodegenerative process is characterized by the progressive ultrastructural alterations of selected classes of neurons accompanied by imbalanced cellular homeostasis, a process which culminates, in the later stages, in cell death and the loss of specific neurological functions. Apart from the neuronal cell impairment in selected areas of the central nervous system which characterizes many neurodegenerative diseases (e.g., Alzheimer’s Disease, Parkinson’s Disease, Huntington’s Disease, etc.), some alterations may be found in the early stages including gliosis and the misfolding or unfolding accumulation of proteins. On the other hand, several common pathophysiological mechanisms can be found early in the course of the disease including altered oxidative metabolism, the loss of cross-talk among the cellular organelles and increased neuroinflammation. Thus, antioxidant compounds have been suggested, in recent years, as a potential strategy for preventing or counteracting neuronal cell death and nutraceutical supplementation has been studied in approaching the early phases of neurodegenerative diseases. The present review will deal with the pathophysiological mechanisms underlying the early stages of the neurodegenerative process. In addition, the potential of nutraceutical supplementation in counteracting these diseases will be assessed.

## 1. Introduction

Neurodegenerative diseases are characterized by the progressive loss of selected vulnerable neuronal populations, leading to an impairment of motor and/or cognitive function [1]. In particular, Alzheimer’s disease entails the degeneration of the entorhinal cortex and the loss of the neurons in the hippocampus that control the memory functions [2]. On the other hand, Parkinson’s disease is characterised by the loss of dopaminergic neurons of the so called “Substantia Nigra” which impacts on the extra-pyramidal control of motor activity [3] and amyotrophic lateral sclerosis (ALS) involves the loss of motor neurons affecting muscle control [4]. Finally, Huntington’s disease leads to the loss of striatal neurons influencing involuntary movements [5]. Therefore, the localization of the neuronal damage provides essential information for the purpose of an early identification and characterization of the disease [6]. Today, very little is known about those mechanisms which contribute to the selective vulnerability of these neurons and many studies have investigated their pathophysiology and prevention. Therefore, the ability to monitor and predict the progression of a neurodegenerative disorder is crucial for the development of targeted treatments. Although the mechanisms underlying neurodegenerative diseases are not clear yet and are different for each disease, they can all lead, to different extents, to death and neuronal loss. These mechanisms are:(1)Alteration of the oxidative metabolism;(2)Loss of the cross-talk among the cell organelles;(3)Increased neuroinflammation.

A representative cartoon, which displays these mechanisms, is provided in Figure 1.

Although the contribution of oxidative stress, inflammation and the imbalance occurring in mitochondria and endoplasmic reticulum appear to play a crucial role in the development of the neurodegenerative process, relatively poor information exists on the potential benefit that occurs when antioxidant product supplementation is carried out in patients undergoing neurodegenerative diseases [7]. Several epidemiological studies have suggested that diets rich in antioxidants play an important role in the protection against various diseases. The main sources of these molecules are found in fruits and vegetables and are associated with lower risks of many disease states including cancer, heart disease, hypertension, neurodegenerative diseases and stroke [8,9,10]. This suggests that the supplementation with natural antioxidants in patients undergoing neurodegenerative diseases should potentiate the efficacy of current treatments even in the early stages of the disease.

On the other hand, the use of relatively safe antioxidant compounds found in the diet as a form of treatment in these disorders is attractive but limited by the difficulty in reaching an active concentration in the brain [11]. Thus, the assessment of the emerging pathophysiological mechanisms occurring in the early stages of neurodegenerative diseases could better address pharmacological and nutraceutical interventions to arrange successful strategies against neuronal loss.

The present review article will deal with the potential of using nutraceuticals in neurodegeneration. This will be assessed on the basis of identification, in the early stages of neurodegenerative diseases, of pathophysiological mechanisms which may be counteracted, at this stage, by means of supplementation with natural antioxidants.

## 2. Pathophysiological Mechanisms Occurring in the Early Stages of Neurodegenerative Diseases

### 2.1. Oxidative Metabolism in Neurodegenerative Diseases

In eukaryotic cells, 90% of the reactive oxygen species (ROS) is generated in the mitochondria, although a small percentage can be found in the other cellular compartments, such as cytosol, peroxisomes and the endoplasmic reticulum [12]. Mitochondria produce ROS as the intermediate result of the reduction of molecular oxygen into water within oxidative phosphorylation in order to generate adenosine triphosphate (ATP). More specifically, ROS are characterized by multiple unpaired electrons and this feature makes them highly reactive and unstable. The brain is the organ with the most active oxidative metabolism, since it consists of a high demand for oxygen and energy. The remarkable demand for oxygen is combined with a lower antioxidant capacity and this makes the brain particularly prone to the action of ROS. In addition to reactive oxygen species, there are also reactive nitrogen species: a superoxide anion reacting with nitric oxide (NO) generates another group of very unstable and reactive molecules called peroxynitrite (RNS) [12].

ROS and RNS largely contribute to neuronal degeneration due to the induced alteration of macromolecules such as DNA, RNA, proteins and lipids [13].

The blood–brain barrier (BBB) neuro vascular unit (NVU), made up of neurons, astrocytes, pericytes, microglia and endothelial cells, can counteract the effects of oxidation when the amount of ROS is limited and when the activity of antioxidant enzymes is preserved. Any imbalance in the NVU functioning may alter the brain homeostasis, resulting in high oxidative stress and the impairment of the integrity of the BBB [14,15].

### 2.2. Loss of Cross-talk among the Cell Organelles

The endoplasmic reticulum is an organelle which preserves many functions at the cellular level; more specifically, it is involved in the protein biosynthesis as well as in the protein folding and translocation, post-translational modifications of proteins including glycosylation reactions, the formation of disulphide bridges and the correct chaperone-dependent folding. Furthermore, the endoplasmic reticulum also regulates the homeostasis of the calcium ion and the synthesis of phospholipids. When these functions cannot be preserved, the cell enters into a stressful condition and the endoplasmic reticulum triggers a sequence of reactions making up the unfolded protein response (UPR). Many neurodegenerative diseases involve the accumulation of protein aggregates and the endoplasmic reticulum stress triggers the activation of an adaptive reaction that restores cellular protein homeostasis, known as proteostasis. Chronic ER stress results in neuronal loss, repressing the synthesis of synaptic proteins [16]. If the stress of the reticulum is not resolved immediately, the UPR can start an autophagic process [17]. Autophagy is a dynamic process culminating with the self-digestion of the damaged cellular elements by autophagosomes, which digest their contents [18]. The UPR and the autophagic process are two cytoprotective mechanisms: their failure activates apoptosis and the cross-talk between the endoplasmic reticulum and mitochondria turns into an exchange of messages leading to cytotoxicity and cell death. More specifically, the calcium ion exits from the lumen of the endoplasmic reticulum. Any change in the membrane potential may open the mitochondrial permeability transition pore (MPTP), with the consequent build-up of the calcium ion in the mitochondrial matrix. The opening of the mitochondrial pore includes the exit of cytochrome c and the activation of caspases [19]. Many neurodegenerative diseases are associated, at early stages, with the endoplasmic reticulum stress as well as with the UPR reactions.

### 2.3. Increased Neuroinflammation

An inflammatory process is started by the cells of the immune system, which activate an inflammatory cascade to protect the body from any damage. However, under some pathological conditions, inflammation is present even in the absence of any specific damage. Recently, a close correlation has been highlighted between inflammation and neurodegenerative diseases. In some cases, in fact, the chronic activation of innate immunity in the central nervous system (CNS) is triggered while, in other cases, peripheral immune cells pass through the BBB, as happens with multiple sclerosis [20].

Microglia and astrocytes are the first cell lines that trigger an immune response by promoting the expression of pro-inflammatory cytokines leading to neurodegeneration, the phagocytosis of synapses and a lower neuronal function [21]. Tumour necrosis factor-alpha (TNFα), interferon gamma (IFNγ) and IL-6 are the main inflammatory cytokines secreted, not only by microglia, but also by the endothelial brain cells and neurons, which infiltrate the immune cells. The release of TNFα activates the transcription factors through mitogen-activated protein kinases (MAPKs) such as ERK, JNK and p38. The transcription factors thus expressed cause the up-regulation of the pro-inflammatory pathways, such as the one connected with the nuclear factor kappa-light-chain-enhancer of the activated B cells (NF-κB) [22].

Oligodendrocytes and macrophages are also involved in the CNS immune responses; in particular, oligodendrocytes are involved in regenerative processes as well as in the replacement of damaged myelin. There is a specific cross-talk among oligodendrocytes and microglia, which is able to select pro-inflammatory responses [23]. The role played by B and T cells is particularly important for neurodegenerative diseases: their functions are altered or antibodies and auto-antibodies are released into the blood, the cerebrospinal fluid and the brain tissues [24]. The involvement of the innate and adaptive immunity in neurodegeneration includes the loss of the BBB integrity. In particular, the following scenario occurs: alteration of the tight junctions that keep the endothelial cells of the NVU together; damage to the endothelium of the NVU; degradation of the NVU endothelium glycocalyx; destruction of glial cells and astrocytes; cell trans-migration and passage of inflammatory lymphocytes through the blood–brain barrier [25].

## 3. Supplementation with Compounds Which Improve the Cellular Antioxidant Effects in Neurodegenerative Diseases

Over time, cells have developed endogenous mechanisms capable of reducing all substances whose accumulation produces cellular alterations.

The molecules able of switching off all or even one of the following pathways can, to various extents, protect the body and slow down the progression of the disease. Some of these compounds represent the golden standard for studies carried out in the early stages of neurodegenerative disorders.

### 3.1. Coenzyme Q10

Coenzyme Q10 (Co-Q10) (a benzoquinone ring with a side chain composed of 10 isoprene units) is synthesized in the body and does not derive from the intake of food. Co-Q10 has important functions, such as the involvement in mitochondrial oxidative phosphorylation for the production of adenosine triphosphate (ATP), the regulation of cell signalling, the modulation of gene expression, a strong antioxidant action and the stabilization of biological membranes [26,27]. Co-Q10 is found in all the body tissues, especially in those featuring a high metabolic activity as well as a significant energy demand; in particular, it is present in the heart, kidney, liver and muscles. Co-Q10 supplementation has proved its ability to minimise cardiovascular damages [28]. Furthermore, Co-Q10 also has strong neuroprotective properties [29]. The mechanisms through which Co-Q10 exercises its neuroprotective effects are associated with the reduction of oxidative stress and apoptotic death [30]. In particular, Co-Q10 increases the mitochondrial function and the formation of ATP and promotes the lipid reduction, thus protecting the body against the build-up of fats due to a unhealthy and unbalanced diet. For example, the administration of Co-Q10 in patients affected by atherosclerosis has led to positive effects on the endothelial function [31]. Moreover, the Co-Q10 supplementation has improved the parameters of those patients who have been subjected to coronary artery bypass surgery or suffer from other cardiovascular diseases [32]. Recently, it has been highlighted that, at the endothelial cell level, the lectin-like oxidized low-density lipoprotein receptor (the LOX-1 receptor) is able to oxidize low-density lipoproteins (oxLDL) and that Co-Q10 may reduce the oxidative stress through the LOX1 modulation [33,34]. In general, the administration of Co-Q10 reduces the oxidative stress, but also inflammation both in vitro and in vivo [35]. In particular, it has been proved that the antioxidant action is performed through the inhibition of the NF-κB transcription factor, whose activation is often involved in the onset of inflammatory processes [36] at the BBB level. Co-Q10 is able to inhibit the release of pro-inflammatory cytokines (IL-6, IL8 and TNF-α) by the endothelial cells as well as to reduce the expression of those proteins (integrins, selectins, ICAM, VCAM) capable of putting the endothelium in contact with the blood monocytes. The reduced expression of these proteins minimises the passage of the immune cells through the BBB [15,37]. Co-Q10 can also reduce the stress of the endoplasmic reticulum organelle, which underlies the main neurodegenerative diseases due to a strong “protein folding” process, the altered communication with the mitochondrion organelles—resulting in oxidative stress—as well as to changes in the concentration of the calcium ion. More specifically, Co-Q10 reduces the expression of the genes relating to endoplasmic reticulum stress, such as those which encode the spliced-X-box binding protein 1 (s-XBP1) transcription factor, calreticulin—the protein that binds to calcium—and the binding immunoglobulin protein “molecular chaperone” [38]. More recently, experimental evidence has proved the direct involvement of Co-Q10 in the expression of a hundred genes responsible for cellular metabolism, the transport of nutrients and signalling. In cases of ataxia, long-term Co-Q10 oral supplementation (30 mg/kg per day) can improve the gait and posture of the patients affected [11].

### 3.2. Glutathione

Glutathione (GSH) is a tripeptide (glycine–glutamine–cysteine) with a thiol group, which preserves the cellular redox state through detoxification reactions. GSH is used by cells to defend themselves against oxidative stress through its GSH–peroxidase and GSH–reductase enzymes. The body reproduces GSH through the recycling of its amino acids and the intake of food rich in sulphur [39]. The excessive outflow of GSH from the cell may alter the redox balance and speed up the cell death by apoptosis [40]. GSH in neurons is highly compartmentalized, but the quantity produced by astrocytes and carried to neurons is higher. In general, the concentration of GSH in the brain amounts to 1–3 mM. The reduction of GSH as a result of mitochondrial structural alterations leads to the accumulation of ROS, a lower functioning of complex I, a higher permeabilization of the outer mitochondrial membrane, the release of cytochrome C and the activation of the apoptotic process. In the nervous system, the activity carried out by GSH and its enzymes is particularly important considering the high sensitivity of neurons to oxidative stress. Therefore, the supplementation or preservation of physiological GSH acts in a neuroprotective way, thus suggesting that an increase in the available GSH pool can be regarded as a promising therapeutic target for neurodegeneration [41]. Neuronal detoxification requires a series of reactions involving GSH [42]. As supported by evidence, GSH not only directly reacts with ROS, but also triggers the activation of the “Nuclear factor erythroid 2-related factor 2” (Nrf2) protein, a transcription factor that regulates the expression of anti-protein oxidants as well as detoxification enzymes. GSH also promotes the reduction of IL-1, IL-6, TNF-α cytokines, together with the caspase-3 activity and the cytosolic cytochrome C levels [43]. The protective effect of GSH is observed also at the BBB level: in fact, the tripeptide has proved its ability to prevent the destruction of the ZO-1 protein, occludins and claudine-5 [44]. The reduction of GSH may lead to metabolic alterations and stress of the endoplasmic reticulum; the latter can cause mitochondrial damages and oxidative stress. More specifically, one of the three UPR branches (IRE1α) interacts with the Bak and Bax pro-apoptotic proteins belonging to the Bcl-2 family. Thus, an altered cross-talk between the endoplasmic reticulum and a mitochondrion leads to cell death by apoptosis. Simultaneously, the exit of calcium from the endoplasmic reticulum may contribute to mitochondrial damages as well as to a further increase in the ROS accumulation [45]. Some neurodegenerative diseases have been associated with disorders in the metabolism of GSH. Clinical studies resulting from nuclear magnetic resonance imaging (NMR) showed lower GSH levels in the brains of patients affected by Alzheimer’s disease (AD) over the control ones. Together with GSH, the activity of the GSH-S-transferase (GST) enzyme, the superoxide dismutase (SOD) enzyme and the GSH/GSSG (GSH/oxidized GSH) ratio were also reduced [46]. Parkinson’s disease (PD) also entails the direct involvement of the GSH levels: a drop of the GSH levels in the brain can be marked as an early event for the diagnosis of PD. Moreover, the reduction of the GSH levels and higher oxidative stress promote the aggregation of the α-synuclein protein [47]. According to some studies, that GSH supplementation led to the dissolution of the α-synuclein aggregates as well as to an increase in the GSH levels in the brains of rats with induced PD [48]. Clinical studies based on NMR proved the involvement of GSH in multiple sclerosis. In particular, the GSH levels in the brains of these patients were lower and GSH supplementation reduced the oxidative stress [49]. GSH is also involved in amyotrophic lateral sclerosis (ALS); GSH levels were reduced in the spine of the affected patients and the altered metabolism of GSH may be regarded as a risk factor in ALS. A recent study highlighted the positive effect of riluzole (a drug currently used to treat ALS) on the synthesis of GSH in glial cells [50].

### 3.3. Vitamin E

Vitamin E is a fat-soluble antioxidant that protects the biological membranes from oxidation and regulates many enzymes responsible for the reduction of ROS/RNS build-ups. The main sources of vitamin E are vegetable oils and foods rich in vegetable oils. Vitamin E includes four isoforms of tocopherols (α, β, γ and δ) and four tocotrienols (α, β, γ and δ). These isoforms cannot be mutually converted and only α-tocopherol meets the need for vitamin E of the human body [51]. The antioxidant property of vitamin E is attributed to the hydroxyl group of the aromatic ring, which gives up one hydrogen atom thus neutralizing radicals or reactive species. Vitamin E proved to be particularly efficient in all those neurodegenerative disorders caused or accompanied by an excess of cellular glutamate. More specifically, nanomolar concentrations of α-tocotriene could prevent the glutamate excitotoxicity in neuronal cells and astrocytes [52]. Recent studies have shown that the administration of vitamin E may have beneficial effects on PD. In particular, an in vivo model of PD was used where mice had been subjected to the interruption of the bidirectional plasticity of cortico-striatal synapses. The intake of α-tocopherol could reverse the abnormality of the synaptic plasticity. These results were not reproduced with any other antioxidant compounds available in food (β-carotene, lycopene, lutein, vitamin C, vitamin A and vitamin K), therefore vitamin E was the specific antioxidant in this experimental model [53]. Recent studies have pointed out that a vitamin E deficiency may compromise the integrity and function of the BBB (blood–brain barrier). In fact, rats subjected to a diet poor in vitamin E had a greater amount of peroxylipids compared to animals provided with the adequate amounts of vitamin E. Eventually, the administration of vitamin E reduced the oxidative stress and alteration of endothelium permeability at the BBB level, thus protecting some tight junction proteins [54].

### 3.4. Polyunsaturated Fatty Acids (PUFA)

Numerous studies have shown that polyunsaturated fatty acids (PUFAs) are involved in neuronal development and growth, therefore their possible therapeutic effects on neurodegenerative diseases are being tested [55]. As previously mentioned, neurodegenerative diseases are associated with increased neuroinflammation but, in many studies, a decrease in brain docosahexaenoic acid (22:6 n-3; DHA) content is also reported. In fact, DHA and arachidonic acid (20:4 n-6; ARA) are the two main PUFAs in the brain and their ratio is known to represent one of the main factors responsible for brain inflammation. The risk of brain disorders has been shown to be higher following a reduced n-3 PUFA plasma concentration and a consequent increase in the ratio between n-6 and n-3 PUFA [56]. The function of PUFAs in the brain is to regulate membrane fluidity (DHA represents 60% of PUFA in neuronal membranes), neuronal survival and signal transduction [57]. Furthermore, the n-3 PUFA, DHA and eicosapentaenoic acid (20:5 n-3; EPA) are precursors of anti-inflammatory resolvins and have an anti-oxidative role as they regulate the activity of proteins involved in oxidative stress in the CNS. Numerous studies have also shown that PUFAs regulate gene expression in the brain [55]. Therefore, epidemiological studies have highlighted a direct association between reducing the risk of inflammatory brain diseases and increasing fish intake. To date, many studies have been conducted to understand the pathology and potential treatments for multiple sclerosis (MS) using the experimental autoimmune encephalomyelite (EAE) animal model. In particular, the role of PUFAs in the prevention and treatment of myelin oligodendrocyte glycoprotein (MOG)-induced EAE has been studied. Studies have been conducted on C57BL/6 female mice fed a control diet and a diet enriched with purified EPA or with the ethyl ester of EPA, diets integrated with a triacylglycerol (TAG) form of DHA or with the hydroxyproline conjugate form. The results of these animal studies fed with n-3 PUFA show that EPA and DHA can delay the onset and progression of disease and reduce its severity. Pre-treatment with DHA or EPA has proven to be particularly effective for the onset of EAE severity. Greater attention is paid to DHA as it is the main n-3 PUFA in neural tissues and since the brain is unable to synthesize DHA from the shorter chain n-3 PUFA, it must be constantly present in the circulation. Its availability in the blood depends on the diet or its synthesis at the level of the liver and adipose tissue (AT) and on the ability to be transported across the BBB. DHA can pass across the BBB because the brain has the capacity to take up plasma un-esterified PUFAs bound to albumin [56,57], in particular DHA, are fundamental constituents of brain cell membranes, at the level of which they esterify to phospholipids together with a saturated chain to form a hybrid lipid. Hybrid lipids are localized at the lipid rafts level in cholesterol-rich membrane domains in which G protein-coupled receptors (GPCRs) and cognate G proteins are present, fundamental in cell signalling. It has been shown that, in disorders such as AD and PD, raft PUFA levels are reduced resulting in the functional alteration of GPCR with a consequent reduction in protein–protein interactions. Therefore, DHA is promising in the treatment of neurodegenerative diseases as lipids containing DHA could prevent neurodegenerative diseases by influencing lipid–protein interactions at the lipid rafts level [58]. Furthermore, in vivo and in vitro studies have reported the involvement of PUFAs in increasing neuronal differentiation and neurite outgrowth. In addition, PUFAs can prevent the neuronal damage associated with neurodegenerative diseases as they improve synaptic plasticity and the formation of new synaptic connections [55]. It is reported that in AD, n-3 PUFAs participate in the process of reducing and resolving inflammation; in particular EPA improves mood disorders, while DHA guarantees the correct brain structure. The positive effect of EPA and DHA is related to the reduction of amyloid load and tau hyperphosphorylation. It is known that the incidence of AD decreases in populations that consume a high amount of fish. PUFAs probably do not act in the advanced stages of the disease, when neuronal impairment is high, but they can be effective in the early stages of the disease, i.e., in the prevention of AD. In fact, studies on AD animal models have shown that dietary supplementation with PUFAs reduces Aβ deposition, improves cognition and reduces hippocampal neuronal loss or neurodegeneration. Studies have also been carried out on the active metabolites of EPA and DHA, such as the oxylipins, demonstrating their anti-inflammatory effects but above all their pro-resolving effects, i.e., their involvement in the resolution of the inflammatory process that does not seem to be regulated in AD [59,60]. Other studies have shown that mitochondrial dysfunction plays an important role in the pathogenesis of neurodegenerative diseases, in particular changes in the concentration and efficiency of the components of the respiratory complexes were reported. The most serious mitochondrial dysfunction in these pathologies seems to be the one that determines the loss of the mitochondrial membrane potential (MMP), closely related to the onset of the apoptotic process. In vivo and in vitro studies have reported the role of PUFAs, in particular DHA, in mitochondrial biogenesis and in the regulation of the genes involved in the brain oxidative metabolism. In particular, the mitochondria of eukaryotic cells are rich in DHA-phospholipids indicating the important role of DHA for the mitochondrial oxidative phosphorylation system (OXPHOS). In fact, the oxidative damage to OXPHOS and the reduced PUFA presence in the mitochondrial membranes cause the MMP loss. Preclinical studies on AD models have recorded an improvement in mitochondrial function after PUFA treatment, reporting their positive effect on ROS production, JNK activation, cytochrome c release and caspase-3 activation. Probably, PUFAs act through action on nuclear receptors, such as PPAR-α, capable of mediating the nuclear effects of EPA and DHA [61]. Many epidemiological and preclinical studies have also highlighted how diets enriched in n-3 PUFA can reduce the risk of onset PD. The studies were conducted in mice treated with 1-methyl-4-phenyl-1,2,3,6-tetrahydropyridin (MPTP), a compound used for the reproduction of the dopaminergic associated features of PD. The presence of EPA and DHA in the diet prevents nigral toxicity caused by MPTP. Other in vivo studies have also reported a partial recovery of the dopaminergic system in the presence of DHA, indicating both a neuroprotective and neurorestorative capacity. Therefore, DHA contributes to the survival of cells in PD as it is involved in the neurotrophic factor secretion pathways, in the inhibition of inflammatory processes, prevents oxidative damage and stimulates cell survival pathways. Furthermore, numerous studies have shown that DHA can interact specifically with the α-synuclein protein, modifying its three-dimensional structure, its oxidant scavenger activity and its propensity to form aggregates and Lewy bodies. Thus, DHA could reduce the neuroinflammation, mitochondrial dysfunction and the oxidative stress generated by α-synuclein alterations, reducing the neuronal degeneration in PD [62,63,64]. These studies have shown that PUFAs could be used for preventive purposes and also as an adjuvant for a better response to therapies in neurodegenerative diseases.

### 3.5. N-acetylcysteine

N-acetylcysteine (NAC) is a precursor of cysteine which promotes the production of the endogenous antioxidant GSH responsible for the balance of the cellular redox state [65]. The bioavailability of NAC is not physiologically high in the brain due to its poor lipophilicity which limits its penetration through the BBB. The replacement of the carboxyl group of NAC with an amide leads to the formation of N-acetylcysteine-amide (NACA), which is more lipophilic and neuroprotective. The NAC compound has proven antioxidant and neuroprotective properties. The antioxidant activity of NAC is expressed through its quick interaction with some reactive radicals such as ·OH, ·NO_2_, CO_3_^−^· and its slower interaction with the superoxide anion (·O^2−^), the hydrogen peroxide (H_2_O_2_) and peroxynitrite (ONOO^−^). NAC is able to modulate many neurotransmitters such as glutamate. The latter is involved in various neurological disorders. In fact, different forms of neuronal damage and degeneration are associated with the excitotoxic damage caused by an altered activation of the N-methyl d-aspartate receptor (NMDAr) [66]. Patients at an early stage of PD show lower GSH levels, ROS accumulation and mitochondrial damage; these factors lead to the specific neuronal death [67]. The administration of NAC results in higher GSH levels in the brain, a reduced oxidative damage and an increase in the synaptic and non-synaptic connections. Moreover, NAC protected neurons from the programmed cell death in rats affected by PD, thanks to the improvement of the functioning of complexes I and IV of oxidative phosphorylation at mitochondrial level [68]. Many studies have shown that the oral administration of NAC interferes with the expression of α-synuclein, the protein that creates insoluble protein aggregates in PD [69]. NAC also has anti-inflammatory properties: in particular, it influences the NF-κB signalling, thus increasing its presence at the cytoplasmatic level and reducing translocation at the nuclear level [70]. At the BBB level, the NAC supplementation avoids losing its integrity thanks to three mechanisms: (a) protection of the tight junctions that keep the endothelial cells together. In particular, a stabilization of claudine-5 and ZO-1 proteins is observed; (b) lowering macropinocytosis; (c) lowering intracellular pores and fenestrations [71]. The protective action of NAC has been largely demonstrated also in AD: more specifically, improvements in learning and memory skills have been observed in the mice affected by the disease, together with an increase in the GSH levels with the following reduction of protein and lipid peroxidation, as increased by the Aβ protein [72]. Again, the NAC supplementation reduced the NF-κB pro-inflammatory activity of neurons and released pro-inflammatory cytokines in astrocytes and microglia [73]. In multiple sclerosis, the NAC supplementation prevented apoptotic death in T cells and minimised the oxidative stress in the CNS in animal models of EAE [74]. Finally, NAC has also proved its effectiveness in ischemia models. The rats affected by ischemia and treated with NAC showed a lower percentage of degenerative neurons compared to the control groups [75].

## 4. Candidate Nutraceuticals for Counteracting Neuronal Cell Death

Supplementation with natural compounds, including plant extracts and nutraceuticals, have recently been studied for their potential in approaching the prevention and/or the management of neurodegenerative disorders, mainly in the early stages of the disease. Here, will be identified the principal candidates which have been studied in recent years.

### 4.1. Polyphenols

Polyphenols are the largest group of phytochemical compounds and in the last twenty years, have been largely investigated by researchers, scientists and nutritionists thanks to their impact on health: polyphenols, in fact, play an important role in degenerative diseases such as cancer, cardiovascular alterations, chronic inflammation and neurodegenerative diseases [76]. Fruits, vegetables, cereals, dried fruit, spices, oil and wine are particularly rich in polyphenols. From a chemical point of view, polyphenols share a common characteristic: the presence of an aromatic ring featuring at least one hydroxyl group. Starting from this basic structure, more than 8000 compounds branch off, based on the number and position of the substituents in the aromatic rings. Experimental evidence has shown that polyphenols can act at the cellular level and minimise the effects of these factors [77]. The main biological activity of polyphenols is the antioxidant one. Depending on the action mechanism, the polyphenolic antioxidants are divided into two groups: (1) primary antioxidants which neutralize the reactive species by giving up an electron or one hydrogen atom [78]; (2) secondary antioxidants, which boost the activity of the antioxidant enzymes such as GSH-peroxidase, catalase, superoxide dismutase and inhibit the expression of pro-oxidant enzymes such as xanthine oxidase [79]. Neuroinflammation is mediated by several factors, such as the NF-κB transcription factor, interleukins (ILs), the tumour necrosis factor alpha (TNF-α), the tumour necrosis factor beta (TNF-β), adhesion molecules, 5-lipoxygenase (5-LOX), 12-lipoxygenase (12-LOX) and 2-cyclooxygenase (COX-2) [80]. As a result of the formation of inflammatory mediators, Toll-like receptors (TLRs) are activated, resulting in the suppression of the genes involved in the inflammatory response. As supported by recent evidence, the anti-inflammatory activity of polyphenols is carried out through the modulation of the expression of TLRs [81]. Other studies highlighted that the modulation of TLRs is induced by polyphenols in inflammatory models featuring neuropathic pain [82]. The protein aggregation of amyloid fibrils (tau, beta-amyloid and α-synuclein) is found in various neurodegenerative diseases and results from the interaction of amino acids at the level of aromatic residues. Polyphenols influence the combination of fibrils, thus weakening the points that cross-interact [83,84]. Some divalent metals, such as copper, zinc and iron, when present in the cell in excessive amounts, may be involved in the ROS production. The intake of polyphenols may chelate these metals under physiological pH conditions [85]. Eventually, mitochondrial dysfunction can be reduced by polyphenols through various processes; these compounds, in fact, regulate the homeostasis of the calcium ion, keep the membrane potential within physiological values and promote the release of cytochrome c into the cytosol when the cell should undergo apoptosis [86]. In neurological pathologies these compounds probably do not act in the advanced stages of the disease, when neuronal impairment is high, but they can be effective in the early stages. However, it is important to highlight that the excessive intake of polyphenols can lead to damage in maintaining health. In fact, it has also been proved that the effects of polyphenols can change depending on the consumed quantity and their bioavailability: in particular, the potential beneficial roles polyphenols have compared to the toxic effects due to their accumulation [87]. To summarise, we can say that polyphenols behave like any drug, showing parallel effects if they are taken in small or large quantities [87].

Among the countless polyphenols, we will investigate two plants containing many polyphenols (turmeric and blueberry) and the use of a phenolic compound resveratrol.

### 4.2. Turmeric

Turmeric comes from the rhizome of the *Curcuma longa L.* plant and belongs to the Zingiberaceae family. It consists of essential oils, polyphenols, carbohydrates, proteins, fats, minerals and water. The main component of the turmeric plant is curcumin, a polyphenolic compound with strong antioxidant, anti-inflammatory, antiviral and antibacterial properties; recently, some studies have proved its potential in neurodegenerative diseases [87]. For example, some studies in vitro on microglia stressed that curcumin, administered in low doses (0–20 μM), promotes the expression of the Heme Oxygenase (HO)-1 antioxidant proteins and peroxiredoxin 6 (Prdx6), minimises neuroinflammation and the production of cytokines (IL-1β, IL-6, TNFα) induced by lipopolysaccharide (LPS) [88], reduces the stress of the endoplasmic reticulum through the inhibition of UPR as well as of the pro-apoptotic pathway associated with the C/EBP homologous protein (CHOP) transcription factor [89], which, once activated, induces the secretion of IL-1β and caspase 1. As supported by evidence, curcumin prevents the brain stress induced by oxidative damage by increasing the GSH levels and the activity of the superoxide dismutase (SOD), GSH-peroxidase (GPx), GSH-reductase (GR) and catalase (CAT) antioxidant enzymes [90]. The anti-inflammatory property of curcumin manifests itself by increasing the levels of anti-inflammatory cytokines as well as the expression of the NF-κB transcription factor [91]. Although curcumin promotes the aforementioned activities and has strong neuroprotective compound, its use is limited by its poor absorption, rapid metabolism with systemic excretion and a limited permeability at the BBB level [92]. These aspects are limitations, but may turn out to be useful. For example, curcumin plays an important role in AD, since it binds Aβ plaques. In fact, the yellow shade of turmeric emits a strong fluorescent signal thus facilitating the diagnosis of AD, but the quick elimination of plaque–curcumin complexes may reduce the extent of the disease [93]. Modern nanotechnologies constantly develop materials able to interact with biological systems, inducing the desired physiological responses and limiting the undesired side effects. Therefore, nanotechnology can influence the ability of drugs to cross the biological barriers and curcumin might be a promising remedy, to be taken through nanocarriers, for the treatment of neurodegenerative diseases [94].

### 4.3. Resveratrol

Resveratrol is a polyphenolic compound present in fruits (grapes, mulberries), roots, cereals, seeds, flowers, vegetables, tea (green and black tea), peanuts and above all, in wine. The presence of high concentrations of resveratrol in wine and its positive effects were confirmed by the “French Paradox” which states that the moderate daily consumption of red wine may protect the body against the increase in triglycerides and cholesterol levels. Many scientific studies have emphasized the important therapeutic effects of resveratrol, focusing on its antioxidant, anti-inflammatory, anti-tumour and cardio-protective properties. Precisely for this reason, the intake of resveratrol is recommended for many diseases [95]. The elements that connect resveratrol to neurodegenerative diseases are increasingly known. The therapeutic effects of resveratrol can be associated with the high antioxidant activity. In Alzheimer’s disease, for example, resveratrol reduces the ROS build-up by increasing the GSH levels; since the cognitive impairment observed in patients affected by AD is related to the ROS amount, the administration of resveratrol may improve this symptomatology. Moreover, resveratrol reduced the levels of nitrite and malonidialdehyde in rats affected by AD [96]. The role played by the antioxidant activity of resveratrol is also proven by the inhibition of the activation of NF-κB of the apoptotic process [97]. Recently, experimental evidence has highlighted the ability of resveratrol to inhibit the aggregation of amyloid fibres thanks to a bond between resveratrol and the N-terminal group of Aβ proteins [98]. As proven by both in vitro and in vivo experiments, resveratrol reduces the mitochondrial dysfunction in Parkinson’s disease [99]. Apart from its antioxidant activity, resveratrol has strong anti-inflammatory properties. According to various studies, resveratrol also inhibits the expression and secretion of many cytokines and molecules with anti-inflammatory function such as nitric oxide (NO), IL-6, TNF-α, TGF-β prostaglandine E2 [96]. Experimental evidence pointed out the ability of resveratrol to activate a series of pro-authaphagic pathways capable of reducing the inflammatory state. The only weakness of resveratrol is its limited bio-availability: in fact, after its intake, it is metabolized within two hours in the liver and intestines and is promptly disposed of. Furthermore, resveratrol has poor water solubility and chemical instability associated with the quick changes in pH or light. Since it takes a long time to treat many neurodegenerative diseases with resveratrol, its biological and pharmacological benefits are reduced [100]. Due to its poor bioavailability, resveratrol barely crosses the BBB, thus reducing its neuroprotective potential. In order to improve the bioavailability of this polyphenol, nano-particles are used which consist of an oil-water emulsion and are solid at room temperature and at the physiological body temperature (37°). The “core” of these nano-particles, which is made up of hydrophobic lipids, is loaded with the substance to be conveyed, which is freely dissolved or dispersed. Thanks to their small size (between 40 and 200 nm), the nano-particles can bio-pass the liver and spleen, thus easily reaching the BBB endothelial cells [101,102].

### 4.4. Ericacee Family

Ericaceae are a plant family with a high content of polyphenols and strong antioxidant properties. The fruits present in our diet are blueberries, cranberries and bearberries. According to experimental evidence, the intake of these fruits has protective effects on the CNS [103]. A study in vivo conducted on rats showed that the administration of cranberries at an amount equal to 2% for 8 weeks led to a significant improvement in the motor coordination and memory of the older animals [104]. Another study analysed in vitro the neuroprotective effects on a model featuring a high build-up of β-amyloid protein. In particular, blueberries can prevent the mitochondrial damages associated with Aβ and reduce the accumulation and aggregation of Aβ through the regulation of NF-κB [105]. A study in vivo conducted on rats affected by neuromuscular disorders, where the animals had been provided with cranberry juice for eight weeks, showed better results for the Rotarod test on motor coordination, compared to the control group. Moreover, the learning ability turned out to be better among the blueberry-fed animals over the placebo group [106]. The intake of blueberries also interferes with the endoplasmic reticulum stress. A study highlighted how the intake of blueberry could modulate the factors associated with the stress of the endoplasmic reticulum through the NF-κB pathway. In particular, the reduction in the expression of UPR-related proteins, such as PERK, GRP78, CHOP, u-XBP1 and IRE1α has been observed [107]. However, today the amount of blueberry that needs to be taken in order to reverse neurodegeneration is not clear yet, alongside the right concentration of blueberry to be taken daily to benefit from the positive effects and avoid the toxic ones.

A representative cartoon which displays the protective mechanisms (oxidative balance maintenance, guaranteed communication between organelles, reduction of inflammation, integrity of the barrier maintenance) is provided in Figure 2.

## 5. Conclusions

The incidence of neurodegenerative disorders (Alzheimer’s disease, Parkinson’s disease, Huntington’s disease, Multiple Sclerosis, Amyotrophic Lateral Sclerosis, etc.) is increasing exponentially. Early stages of neurodegenerative process are characterized by common mechanisms: oxidative stress, loss of cross-talk among the cell organelles and increased neuroinflammation culminating in cell death. A correct maintenance of the endogenous antioxidants and supplementation with natural nutraceuticals have been suggested to counteract the oxidative damage and inflammation in brain tissues occurring in patients undergoing neurodegenerative processes. Since it is known that the accumulation of all these compounds can cause toxicity, further clinical trials are required to support an extensive use of the right quantities to hire/produce, mainly in the early stages of neurodegenerative diseases.

## Figures and Tables

**Figure 1 ijms-21-02618-f001:**
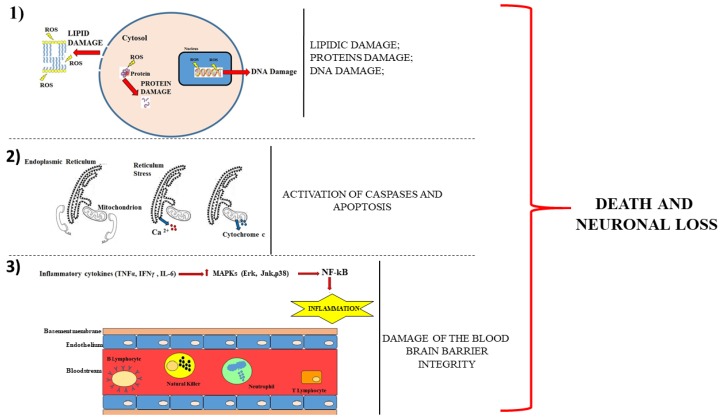
Description of the main mechanisms involved in neuronal damage. The panel (**1**) shows that the cell death occurs following the damage to proteins, lipids and DNA. Panel (**2**) describes how the loss of communication between the endoplasmic reticulum and the mitochondrion leads to the leakage of calcium ion from the reticulum and of cytochrome c from the mitochondrion. The consequences are caspases activation and apoptotic cell death. Panel (**3**) shows a section of the blood–brain barrier with the endothelium surrounding the bloodstream. The production of cytokines determines the activation of mitogen-activated protein kinases (MAPKs), which increase the immune cells in the blood. The breaking of the integrity of the barrier favours the infiltration of these cells encouraging neurodegeneration.

**Figure 2 ijms-21-02618-f002:**
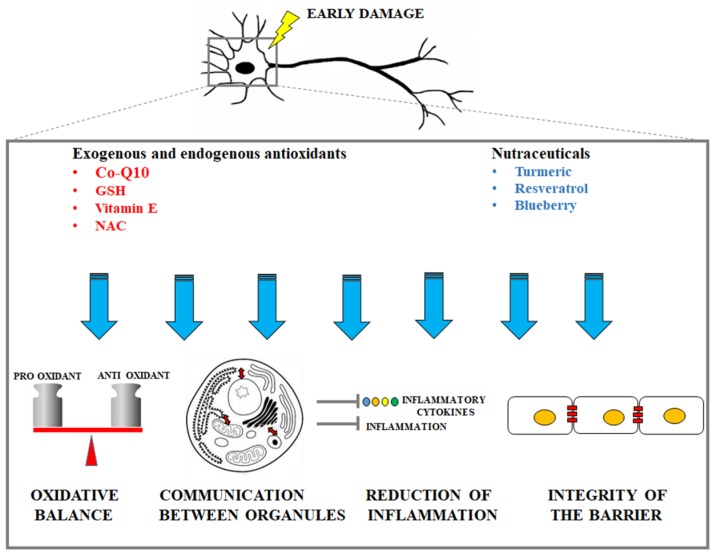
Description of the main protective mechanisms by exogenous antioxidants and nutraceuticals in early neuronal damage.

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
