# Peer review of "The Potential for Natural Antioxidant Supplementation in the Early Stages of Neurodegenerative Disorders"

_ijms, 2020, doi:10.3390/ijms21072618_

Round 1

Reviewer 1 Report

The work is a complete and well-structured review of the common molecular mechanisms underlying neurodegenerative diseases and the effects of different antioxidants. Although it is a topic for which there are numerous reviews in the literature, not many study the different neurodegenerative diseases together.
In my opinion, figure 1 should be integrated into a single figure, interrelated the mechanisms involved in neuronal damage.

The first part of the conclusion should be reformulated because it refers to something that is not commented on in this work.

Minor review:
TNF alpha first appears on line 125, however its abbreviation is defined on line 172.

The following sentence must be corrected: The reduction of the expression of ...(Line 174)

Author Response

Reviewer 1

Dear reviewer,

thank you for your corrections and your suggestions. My answers below are highlighted in yellow and the revision in the text of the paper is also indicated by a vertical red line.

Comments and Suggestions for Authors

Major review:

The work is a complete and well-structured review of the common molecular mechanisms underlying neurodegenerative diseases and the effects of different antioxidants. Although it is a topic for which there are numerous reviews in the literature, not many study the different neurodegenerative diseases together.

1) In my opinion, figure 1 should be integrated into a single figure, interrelated the mechanisms involved in neuronal damage.

1) Figure 1 has been modified. It was not possible to describe all the panels in a single figure because in every panel a different mechanism is described. Nevertheless, I have correlated the three mechanisms in a common consequence (as indicated in parentheses) which is death and neural loss  I added a summary sentence in the text (lines 49-50 51). Additionally an explanatory caption has been added (lies 59-65).

2) The first part of the conclusion should be reformulated because it refers to something that is not commented on in this work.

2) The first part of the conclusion has been reformulated (lines .522-529).

Minor review:

1) TNF alpha first appears on line 125, however its abbreviation is defined on line 172.

1) Tumour necrosis factor-alpha (TNFα) was defined the first time (line 137) and subsequently abbreviation has been used.

2) The following sentence must be corrected: The reduction of the expression of ...(Line 174).

2) The sentence “The reduction of the expression of…(line 189) has been corrected (Line 189)

Reviewer 2 Report

The paper is reviewing the recent findings on the field of treatment of early stages of neurodegenerative disorders. However, the title is not completely correct, since not all of the substances presented in the paper are antioxidants (e.g. PUFA, N-acetylcysteine). The chapter 3 should deal with exogenous antioxidants, but there are some that are endogenous. So authors should rethink the title of the section 3.

There is quite a lot of mess with abbreviations, so authors should decide which ones to use, define them on the first time and then use them all the time. Some of abbreviations are not necessary (see also comments below).

The authors could also check for the newest papers from 202, since there are already several (also review papers) published on the same field.

More specific comments: 

Replace “anti-oxidant” replace with antioxidant throughout the document

Avoid using abbreviations that only appear once or twice in the text (e.g. CRT, BIP…)

Section 3: “Supplementation with exogenous antioxidants” – Co-Q10 and GSH are endogenous

Vitamins should not be written in capital.

Line 63: “fishes, fruits and vegetables “ - fishes are not one of the main sources of antioxidants

Lin 64-66: why only the patients with neurodegenerative diseases, what about the prevention before the onset of disease?

Line 80: “reticulum. [12].” – remove the extra dot before the reference

Line 91: define BBB

Line 130 “NFkB” – replace with the correct abbreviation: NF-κB, change also in the Figure 1

Figure 1-2: the abbreviation ER is not necessary

Figure 1-2: show also “activation of caspases”, which is mentioned in the text

Figure 1-3: “ERK, JNK e p38” – replace by ERK, JNK and p38

Figure 1-3 – lower scheme is not clear and not explained in the text well enough (lines 137-141)

Line 169: “NF-kB” – replace with NF-κB and check the rest of document

Line 171: define BBB the first time it appears (line 91), then use the abbreviation throughout the document

Line 226: NMR is already defined in lane 216

Lines 221, 227, 228,…: use abbreviation for glutathione after first explanation

Line 236: “found in food” - should be removed

Line 254: Define the abbreviation

Line 267: abbreviation for CNS should be used

Line 270: define MS, EAE

Line 300: Define AD when it first appears. Same for PD. Or do not use abbreviations at all.

Line 343: Correct H2O2

Line 346: define NMDA

Line 353: use α-synuclein through all document (upper you use αSyn)

Lines 382-385: the mechanisms of neurodegeneration is well explained already before, no need to do it again

Line 411: turmeric and blueberry are not a polyphenol, they contain several of them…

Lines 414-415: “curcumin which consists of essential oils, polyphenols, 414 carbohydrates, proteins, fats, minerals and water.” – not true, Curcumin is a polyphenol

Line 419-422: define HO-1, LPS, UPR, CHOP

Line 439 “fruit” – use plural + peanurs are not fruits

Line 443-445: missing references

Line 474: Not correct title, since all the berries are included in the chapter

Line 493: Which protective mechanisms, name them

Figure 2: In the text, there is nothing about pro-oxidant properties of these substances; “organules” – replace by organelles

Conclusions: The possible toxical effects should be mentioned (see DOI: 10.3390/antiox9010061)

Round 2

Reviewer 2 Report

The manuscript has been considerably improved.

Please correct radicals - line 361 ( OH, NO2,CO3–·).

Author Response

Reviewer 2

Minor Ravisions

1) Please correct radicals - line 361 ( OH, NO2,CO3–·).

1) Radicals have been corrected (Line 353).